# Testing the Accuracy of Wearable Technology to Assess Sleep Behaviour in Domestic Dogs: A Prospective Tool for Animal Welfare Assessment in Kennels

**DOI:** 10.3390/ani13091467

**Published:** 2023-04-26

**Authors:** Ivana Gabriela Schork, Isabele Aparecida Manzo, Marcos Roberto Beiral de Oliveira, Fernanda Vieira Costa, Robert John Young, Cristiano Schetini De Azevedo

**Affiliations:** 1School of Sciences, Engineering & Environment, Peel Building, University of Salford, Manchester M5 4WT, UK; ivanaschork@gmail.com (I.G.S.); r.j.young@salford.ac.uk (R.J.Y.); 2Departamento de Evolução, Biodiversidade e Meio Ambiente, Instituto de Ciências Exatas e Biológicas, Universidade Federal de Ouro Preto, Campus Morro do Cruzeiro, s/n, Bauxita, Ouro Preto 35400-000, Minas Gerais, Brazil; isabelimanzo@gmail.com (I.A.M.); marcosbeiraloliveira@gmail.com (M.R.B.d.O.); 3Departamento de Ecologia, Instituto de Ciências Biológicas, Bloco E, s/n, Universidade de Brasília, Campus Darcy Ribeiro, Asa Norte, Brasília 70910-900, Distrito Federal, Brazil; costa.fvc@gmail.com

**Keywords:** sleep, Petpace^TM^ collar, physiology, wellbeing

## Abstract

**Simple Summary:**

Quality and quantity of sleep can be, potentially, used as an animal welfare indicator. However, sleep data collection can be difficult, since it often involves animal manipulation, which can disrupt sleep patterns. Thus, it is important to test new, non-invasive, methodologies to measure sleep. Wearable technologies are now being tested in sleep studies. However, data recorded by wearable technologies need to be validated by comparisons with data collected using standard methods of behavioural recording. In this paper, we tested the accuracy of wearable technology to investigate sleep behaviour in domestic dogs. To acquire behaviour and physiological data from the dogs, the study used a smart-sensing collar from the brand PetPace™. Behavioural data collected by the collar were compared with data collected using focal sampling with instantaneous recording every 30 s for 20 days. Comparisons between methods showed differences in certain behaviours, such as inactivity and activity for diurnal recordings. Despite this, total activity and total sleep recorded were similar between methods. Overall, the used wearable technology shows potential to be a useful, and a less-time consuming, tool for the evaluation of behaviours and for the assessment of wellbeing in dogs.

**Abstract:**

Sleep is a physiological process that has been shown to impact both physical and psychological heath of individuals when compromised; hence, it has the potential to be used as an indicator of animal welfare. Nonetheless, evaluating sleep in non-human species normally involves manipulation of the subjects (i.e., placement of electrodes on the cranium), and most studies are conducted in a laboratory setting, which limits the generalisability of information obtained, and the species investigated. In this study, we evaluated an alternative method of assessing sleep behaviour in domestic dogs, using a wearable sensor, and compared the measurements obtained to behavioural observations to evaluate accuracy. Differences between methods ranged from 0.13% to 59.3% for diurnal observations and 0.1% to 95.9% for nocturnal observations for point-by-point observations. Comparisons between methods showed significant differences in certain behaviours, such as inactivity and activity for diurnal recordings. However, total activity and total sleep recorded did not differ statistically between methods. Overall, the wearable technology tested was found to be a useful, and a less-time consuming, tool in comparison to direct behavioural observations for the evaluation of behaviours and their indication of wellbeing in dogs. The agreement between the wearable technology and directly observed data ranged from 75% to 99% for recorded behaviours, and these results are similar to previous findings in the literature.

## 1. Introduction

Sleep is a fundamental physiological process across animals, being an intrinsic part of the homeostatic process and an essential behaviour that cannot be eliminated or disrupted without deleterious consequences to animals [1,2,3]. Sleep also is a physiological process that has similar behavioural and physiological components across different mammal species [4,5,6]. Despite this, different species have different characteristics as to how sleep behaviour occurs (e.g., number of bouts, period of the day), although all species have similar sleeping cycles, starting with slow-wave sleep, followed by REM sleep (Rapid Eye Movement) and then, wakefulness [6,7,8].

Sleep is directly affected by the environment and has an important emotional function, since events experienced when the individual is awake are directly connected with subsequent quality of sleep [9,10]. Furthermore, studies have shown that sleep quality and quantity is affected by both acute and chronic stress, and that lack of sleep is a major stressor [11,12]. Therefore, these characteristics indicate that sleep can be a reliable animal welfare indicator, although, to date, little research has been conducted to use sleep as a welfare measure, most likely due to the practical difficulties to measure this behaviour [13,14].

The gold standard method to measure sleep is the use of EEG (electroencephalogram) recordings, which involves placing electrodes on an individual’s head to measure brain waves associated with sleep [15]. In non-human species, most of the time, this means having the electrodes surgically implanted under general anaesthesia and a prolonged post-operative recovery period, a process that can have many health risks, and an overall negative impact on animal welfare [13,16]. An alternative to this procedure would be the use of external adhesive electrodes, a method successfully tested with cows [17,18] and owls [16], but not without drawbacks. For instance, cows sometimes rubbed off the electrodes and in owls, an attenuation of signal was observed over time, which affected data collection. Other problems associated with this method are the size of the electrode which could not be fitted on some species, such as smaller animals (e.g., mice), cost and particularities (e.g., need for specific positioning on the head) of such equipment.

In the past decade, the use of accelerometers to measure different behaviours, and assess behavioural problems has been developed in animal welfare research [19,20,21]. Accelerometers are devices that measure the difference in acceleration forces in relation to the Earth [22,23]. In behaviour research, when affixed to an animal (on their legs, neck or back), one to three of these sensors are aligned such that each one of them identifies acceleration in a single plane (dimension) of movement—surge, heave and sway [24,25]. The three sensors working together can represent, in real time, three-dimensional movement, which provides data on precise changes in behaviour [19,23].

Accelerometers have been used for different purposes, such as monitoring body movement, postures, reproduction, activity budgets, stress levels, inactivity/rest and, most recently, sleep [20,21,26,27,28]. Although accelerometer data alone cannot discriminate between sleep phases, the association of this method with other remote assessed physiological measures, such as heart rate or body temperature [29,30], could make this technology a reliable tool to assess sleep quantity and quality (e.g., sleep fragmentation or reduced sleeping times). Another option would be to associate accelerometer measures with recordings of specific behavioural events that happened during sleep; for example, cows (*Bos taurus*), giraffes (*Giraffa camelopardalis*), elephants (*Loxodonta africana*) and horses (*Equus caballus*) all need to lay down in lateral recumbency to achieve REM sleep [31,32,33,34].

In dogs, *Canis lupus familiaris*, our model species, the use of an accelerometer-based technology has been adapted in the form of wearable collars that can track dogs’ activities, behaviour patterns and even stress-related responses [35,36,37,38,39]. Not only being used for scientific research, but these types of collars were also further developed into a commercial product that owners can acquire for their pet and veterinarians can use to monitor their patients [37,39,40]. Nonetheless, despite the applicability of such collars, only a few studies have evaluated the precision of this technology to predict rest, and whilst results are promising in relation to head posture and low body movement, further work is still necessary to be able to identify the quality of the rest an animal achieves as monitored by these devices [35,41].

Previous studies that used accelerometer-based technology to assess rest in dogs found different efficiencies in the methods. For example, an accuracy of 80–90% between collar and behavioural observations has been found when assessing rest based on head-down recumbency, which is a characteristic of sleep and sustained rest in dogs [35]. However, inaccuracy sometimes happened if the dog’s head was in an inclined angle and, if the dog was more restless during recumbency. Other researchers had a similar problem, as in the previous study, but added a rotation correction step designed to re-orient the sensor data. Thus, small movements were detected, hence a change in angle or sudden small movements did not cause an incorrect score, whilst providing information on head position during sleep [41]. Researchers that used accelerometer-based technology suggested an acceptable level of variation in the accuracy of the measures made by wearable technologies of around 80–100% [42,43,44].

Domestic dogs are a diurnal species, which mean they have a clear diurnal pattern of activity (most behavioural processes happening during the day) [45,46]. Moreover, wakefulness comprises 70% of daytime in comparison to less than 40% in the dark cycle, when sleep related behaviours are predominant [46,47,48]. Dogs are polyphasic sleepers, which mean they frequently transition in an out of sleep multiple times per night, and exhibit sleep (nap) bouts during the day [45,48,49].

The Petpace^TM^ collar (PetPace, LLC, Burlington, MA, USA) is currently the only commercial version of a tri-axial accelerometer to measure dogs’ activity, which has advanced features appropriate for scientific research (such as access to RAW data and live-transmission of a comprehensive data set) [37,40]. However, its accuracy in measuring sleep behaviour in dogs has not been tested. Therefore, this research aimed to investigate the accuracy of this device to monitor sleep and physiology in kennelled dogs and, thereby, determine if this device could be a useful tool to investigate sleep. For this investigation, data collected by the Petpace^TM^ collars were compared to data collected simultaneously by a standard behavioural recording method, to evaluate if there were any significant differences in the data generated by both methods. This aim is derived from the need to measure sleep using methods that are non-invasive in nature.

## 2. Methods

### 2.1. Ethical Statement

This study was submitted to and approved by the Science & Technology Research Ethics Panel of the University of Salford, Manchester (STR1617-80), and by the Commission of Ethical Use of Animals in Research of the Universidade Federal de Ouro Preto, Minas Gerais–Brazil (Protocol 2017/04). The present study was conducted in an established dog population from the above-mentioned facility without any modifications made to their routine or management of kennels. Dog husbandry procedures followed the National Animal experimentation Control Council, Normative Resolution No. 12 [50] and were carried out by the main staff on site.

### 2.2. Study Site and Subjects

From the kennel population of Centre for the Animal Science, in the Federal University of Ouro Preto, state of Minas Gerais, Brazil, six female and seven male adult mixed-breeds dogs (mean age ± SD: 5.8 ± 1.8 years, range 2–7 years old; mean weight 24.04 ± 5.49 kg) were randomly selected for the study. Dogs were kept in same-sex pairs (aside from one male that was housed alone during our research) in outdoor kennels (5.8 m × 1.6 m × 1.65 m) with bare concrete flooring and walls. One-third of the space was covered for shelter and females had an additional small room in the back of their kennels which was used as a birthing den, if the females were selected for breeding. Dogs were not walked but received exercise/play sessions in same-sex groups in a separate area within the facility.

Dogs had access to water and food *ad libitum*, which was replenished as needed and kennels were cleaned twice a day. Dogs did not have contact with other dogs or people outside the designated play sessions and management routines. Additionally, dogs had daily health inspections and throughout the duration of our study were considered clinically healthy and without behavioural problems by the veterinarians responsible for their health. Finally, no dogs took part in another study or were selected for breeding while being part of the present research.

### 2.3. Data Collection

For this study, dogs’ behaviour was assessed using two different methodologies: standard behavioural observations [51] and a smart sensing collar (wearable technology). Observations were conducted in 5-day assessment periods from Monday morning to the following Saturday morning from October 2017 to May 2018. Due to some limitations with equipment (number of CCTV cameras/kennels simultaneously recording were limited to four kennels/eight cameras per observational period), not all dogs were observed simultaneously, but pairs were always accessed together in the same week. At the end of the experiment, we had observed each dog for a total of 20 days and nights in consecutive 5-day assessments (4 weeks per dog), with observation periods spread equally across all dogs (i.e., dogs were observed at the beginning, middle, and end of the experiment to avoid bias with changes in temperature and day-length due to kennels being exposed to ambient light and temperature levels).

### 2.4. Behavioural Observations

This study was conducted in parallel with another investigation and a full description of the methods for behavioural assessment can be found in [52]. In summary, during any assessment week, dogs were observed for 24-h using CCTV cameras with night vision capabilities. The observation period was divided between day (7:00–17:59) and night (18:00–06:59). Diurnal observations sampled behaviour for 15 min in every hour using a focal sampling with instantaneous recordings of behaviour using a 30-s interval [52]. For the nocturnal observations, we were interested in the duration of sleep, therefore this behaviour was quantified using focal animal sampling with continuous recording of behaviour [52]. The behaviours recorded during day and night were allocated into three broad categories: rest/inactivity, sleep, and activity. Behaviours were classified using an ethogram for dogs based on the literature (Supplementary Material Appendix A) [53,54,55].

### 2.5. The Wearable Technology: Behaviour and Physiological Metrics

To acquire behaviour, we used a smart-sensing collar from the brand PetPace™ (PetPace, LLC, Burlington, MA, USA). The PetPace™ is a non-invasive wireless collar that continuously collects a dog’s vital signs (heart rate and respiratory rate) and behaviour patterns, and then transmits the data to an online database using a gateway connected to the internet.

The collars continuously monitored a dog’s activity and body posture through a tri-axial accelerometer. The device has a hard plastic casing, measuring 40 × 35 × 15 mm and weighing 43 g, attached to an adjustable collar. Collars were fastened on each dog such that the activity monitor was located ventrally on the dog’s neck. Each dog had an appropriate collar size (S, M, L) based on the dog’s weight, as instructed by the manufacturer’s guidelines [56].

Using the PetPace online platform, it was possible to create an individual profile for each dog, which was associated with a specific collar; this also enabled us to switch collars between dogs without losing data. At the beginning of the assessment period, each collar was turned on in the laboratory while in range of the internet gateway, to signal to the database that a dog was being monitored. Once the collar showed as active in the dogs’ profile, the collar was then taken to the kennels and fitted to a specific dog. Furthermore, collars were removed once a day, on the following day, and moved back in range of the internet gateway so that the logged data were uploaded to the PetPace online platform. The collars upload data automatically if they are in range of their gateway; however, the kennels facility did not have internet access, creating the need to remove the collars from the dogs. This process took no longer than 30 min and any accelerometer data from this interval were discarded from the analysis. At the end of each assessment week, all data present in the online database for each observation period were downloaded as an Excel file which would be used for analysis.

The behaviours recorded by the collars during day and night were also allocated into three broad categories: rest/inactivity, sleep, and activity. The collar recorded the duration of each behaviour in seconds. For dogs’ behaviours, the accelerometer registered changes in position every second and reported a score with maximum recordings every 2–3 min (this feature is not adjustable). For dogs’ rest/inactivity, the collar recorded six behaviours based on the position of the accelerometer: standing, sitting, lying right, lying left, lying sternally, and lying back. The total duration of each of these behaviours (in seconds) was used in the analysis for the rest/inactivity category. For dogs’ activity, the collar classified dogs’ activity as having low activity, medium activity, or high activity, but without allocating the activity to a specific behaviour (e.g., running or playing), instead the collar measured intensity of movement. The threshold between the different categories, variation from low to medium, for example, is determined by an algorithm, which is not disclosed by the manufacturer.

For sleep, the collar provided information based on the same points threshold system used for activity and automatically generated a sleep score reported as a percentage (e.g., 73% of sleep for 22 November 2017). Hence, it was not possible to account for duration of sleep only using this specific parameter since the sleep average would be based on mean points recorded as sleep, not duration of sleep in seconds. To circumvent this problem, an index of sleep was created based on the duration of the body positions in seconds, which were classified as rest/inactivity by the collar and showed as a zero-point in the behaviour output summary (i.e., no perceivable movement). Thus, the mean in seconds of lying behaviour recorded between 18:00 and 07:00 (same period used for continuous observations of sleeping behaviour) was calculated and divided by the total number of resting/inactivity points recorded by the collar as rest and that were equal to zero in the activity threshold. For example, the collar recorded a mean of 33,815 s of rest for a dog, distributed between 1616 points of recorded rest (in the spreadsheet this appears in a column as the number zero for activity and in the subsequent column as rest). Therefore, the average per point would be the total of seconds divided by the total of points. In the example, 20.92 s per point. The calculation gave us an approximation of duration per total points (sum of duration per points) per night. The same approach was used to verify sleep during the day.

### 2.6. Statistical Analysis

All data acquired by the collars and from the observations were tested for normality using the Anderson–Darling normality test. All statistical tests were considered significant at *p* < 0.05. Descriptive statistics of all the analysed metrics were conducted and results are presented as either mean counts or percentages with standard deviations.

To verify the efficiency of the measures acquired by the collar against the metrics recorded by the video observations, behaviours were separated in three large categories: activity, inactivity, and sleep. From the baseline data, percentages were estimated for each category by method, as well total values for day vs. night observations and results compared using Wilcoxon ranked test for paired measures [57].

To characterize and compare the different categories of behaviours (i.e., sleep, rest/inactivity, low, medium, or high activity), a Wilcoxon ranked test was carried out [48]. To verify difference in behavioural expression between categories of behaviour, a Friedman test with Dunn’s post hoc test was used [57]. Spearman rank correlations were used to verify associations between different physiological measures (i.e., heart and respiration rates) [57]. 

Additionally, to verify if the point-based system of the collar would also be a good metric to compare against the behavioural observations, two categories were created based on the most expressed behaviour for each period of observations: total activity for diurnal recordings and total sleep for night recordings. Once again, variables were compared against each other using Wilcoxon ranked test for paired measures [57].

All statistical analyses were carried using RStudio [58]. Data generated and analysed in this study are available at Mendeley Data Repository website.

## 3. Results

### 3.1. Characteristics of Behaviour Collected by Wearable Technology Collars

At the end of the observation period, data from the PetPace generated 488,800 points of activity for all the dogs. Overall, rest/inactivity was the highest activity recorded by the collar (66.2 ± 2.6%), followed by medium (18.9 ± 19%), low (9.5 ± 1.6%) and high (5.4 ± 2.7%) activity categories. Highest levels of activity were mostly recorded at 09:00 h and at 18:00 h. Activity levels during the day varied, however a steady decrease in activity was observed from 18:00 h onwards and sleep was recorded at similar levels (not statistically different) from 22:00 h until 07:00 h the following morning.

Variation in activity levels were significantly different between periods of the day, as evaluated by the Wilcoxon test. Rest/inactivity was recorded most at night (W = −151.0, *p* = 0.018), while medium (W = 262.0, *p* = 0.007) and low (W = 259.6, *p* = 0.002) activities were most recorded during the day (Figure 1). The expression of high activity did not differ between day and night (Figure 1).

### 3.2. Sleep Metrics Collected by Wearable Technology Collars

Dogs slept 6.6 ± 1.6 h during the night and 57.9% of the total number of rest points scored as zero occurred between 00:00 h and 05:00 h. During the day, sleep was recorded for 0.7 ± 0.4 h and 37.11% of rest points were acquired between 10:00 h and 12:00 h. Sleep was significantly different between periods of day, with dogs sleeping mainly at night (W = −91.0, *p* < 0.001).

### 3.3. Evaluating the Efficiency of the Wearable Technology Collars against Behavioural Observations Made by a Human Observer

When evaluating the methods based on behaviour categories, differences between methods ranged from 0.13% to 59.3% for diurnal observations (mean ± SD: 18.7% ± 11.6%) and 0.1% to 95.9% for nocturnal observations (mean ± SD: 19.8% ± 24.5%). Significant differences were found between the methods (Table 1). The collar registered more activity during the day (W = −7986, *p* < 0.0001) and more sleep at night (W = −8065, *p* < 0.0001), when compared to observations made from the video recordings. Consequently, the collar registered less activity at night (W = 8333, *p* < 0.0001) and less inactivity during the day (W = 8244, *p*< 0.0001), than the behavioural observations (Figure 2). Individual category scores can be found in Table 1.

For the categories activity and sleep using the collar points threshold, differences were found for activity during the day (W = −7986, *p* < 0.0001, Figure 2A), inactivity during the day (W = 8244, *p* < 0.0001, Figure 2A), activity at night (W = 8333, *p* < 0.0001, Figure 2B) and sleep during the night (W = −8065, *p* < 0.0001, Figure 2B) between the methods. However, no difference was found for the total diurnal and nocturnal observations between the methods (when all recordings, regardless of the behaviour, were summed; Figure 2C,D).

Similarly, when one method was compared to the other using absolute values per animal per category, we found no difference for sleep between methods (Figure 3B), but for activity records, we found differences between the collar and the observations (Figure 3A,C).

## 4. Discussion

### 4.1. Characteristics of Behaviour Collected by Wearable Technology Collar

The objective of this study was to ground truth the use of wearable technology (PetPace) collars in assessing activity and sleep. The results suggested that the collar is a valuable tool when assessing dogs’ activity and variation in behaviour, being able to verify even subtle changes that most likely a human observer would not detect. The collar showed the distribution of activity levels throughout the day, and the differences in the amount of activity between day and night, in a similar distribution to direct behavioural observations.

Similar to the results found in [51] for the same population, the collar also verified that the dogs slept most at night but spent most of their time inactive during the day. It also demonstrated that the higher patterns of activity happened in two specific points during the day, at 09:00 and at 18:00. Both are hours of high human movement around the facility due to management practices, such as cleaning the kennels, feeding the dogs and, for the afternoon time, end of the shift for the day, when most staff are leaving. Furthermore, these times also correspond with the beginning and end of the students’ classes in the university for the day, which increases traffic of people and vehicles around the kennels. All these factors could be contributing to arousing the dogs, thereby producing longer bouts of high activity compared to other periods of the day.

Most importantly, the activity points acquired by the collars seemed to be consistent in the way they measured activity, as no errant patterns were verified between days or between the same animals; that is, the collars were precise. The PetPace collar has been tested against a few other accelerometers that have been validated and are commonly used for scientific research in both humans and non-human subjects [37,40]. For the most used brands, Actigraph and Actical, the collar achieved high levels of concordance between the data—84% and 72%, respectively [40]—meaning that it is a reliable tool for measuring activity.

### 4.2. Evaluating the Efficiency of the Wearable Technology Collar against Behavioural Observations

The results produced by the collar showed the same distribution of activity and sleep as found in previous studies [51,59]. However, the amount of time quantified by the collar versus the amount of time reported by the behavioural observers were significantly different when allocated in the three broad categories used for the analysis (ranging between 18% and 25% of difference). Only inactivity was scored statistically similar by the different methods: similarity in data points was found for total percentages without breaking down in specific behavioural categories, but only for nocturnal observations. Despite our attempts to collect similar data from both methodologies, it is likely that a difference still occurred, which may partially explain our results. In summary, data from collars (just like direct behavioural observations) are precise but not necessarily accurate. Our results also mean that care must be taken when comparing data collected by these different methods.

For the nocturnal behaviours, as the data were being scored continuously by the observer and the goal was to identify patterns of sleep, and only three categories were used during the video analysis: active, rest/inactive, and sleeping. Nonetheless, we found similarities between the results of each method. Additionally, the amount of time the behaviour was being recorded manually was significantly less than the recordings made by the collar. Video analysis by a human observer will be less precise in recording data than an electronic sensor. The human eye can detect changes in tenths of a second, whereas an electronic sensor could be in milliseconds; this difference may explain our results.

To improve accuracy between the methods for the diurnal observations, a sample point by sample point analysis of behaviour should probably be used instead of percentages by day. This means that each observation made manually would have to be matched by the exact time from the collar, although this would be a very time-consuming task, as opposed to comparing means for the same days, as was performed for the analysis in this study. Thus, the differences detected could be due to not exactly time-matching observations.

When using the values of the points-based system of the collar converted into a ratio, we were able to find more appropriate results than using the ones from the previous analysis, especially when looking at the mean variation of sleep and activity in individual dogs. As activity and rest were only scored as points, not as duration of behaviour (e.g., sleep duration or locomotion duration), the calculation of the ratio appears to have provided a more realistic result than an activity threshold. When evaluating such parameters in relation to animal welfare, the quantification of total time of a certain behaviour provides more reliable information than a score, but still, if the scoring can be monitored over time individually, the variation in activity levels could be a good alternative to observing several hours of video.

### 4.3. Accuracy of Sleep Parameters Registered by Wearable Technology Collar

The ability of the collar to measure behaviour had contradictory results. From the perspective of collecting rest points and generating an automatic sleep score index, the collar was efficient, especially if the data were transmitted live. Contrarily, as an indicator for sleep efficiency, adjustments are necessary. As for understanding the architecture of dog sleep, the collar should be reporting sleep in duration, as it does with the other behavioural categories. Even though the collar does report different lying position in seconds, sometimes the animal can be inactive, but not necessarily sleeping. Furthermore, it would be ideal to have a threshold that could account for the number of time that the dogs changed from sleep into wakefulness (sleeping bouts). As sleep fragmentation compromises both physiological and psychological health [60,61], this feature could be an indication of increase or decrease in sleep quality when assessing the same individual over multiple nights.

Overall, we observed that the PetPace collar does not appear to have a problem with its rest/inactivity threshold, since no resting points were scored with a value greater than zero. Despite this, the point-based system to score sleep does not seem to be the most appropriate way of quantifying sleep or monitoring changing patterns in this behaviour. As the collar only reports a final quantity of sleep, not the temporal patterning of behaviour, it is impossible to know, for example, the duration of continuous sleeping bouts or the total number of episodes in a night. Thus, two nights may report animals sleeping for the same amount of time, but their sleep profiles could differ significantly.

The main difference between the PetPace and other monitoring devices is the fact that it provides real-time information if connected to a wi-fi network, however there is the possibility of data loss if the collar is out of the range of the internet gateway for more than 12 h. Since the kennels did not have internet connection available, we had to remove the collars from the dogs daily, to download the data, which meant for a period the collar was not acquiring any data. Moreover, because we were not allowed on site after work hours (6 pm), our collars stayed away from the gateway longer than instructed by the manufacturer. Although there was data loss in our study (12% of total recorded behaviour), this did not have a significant effect on our results due to the extensive amount of time the dogs were monitored. However, we highlight that, for studies with smaller data collection periods, loss of data could be a problem.

Finally, it is important to comment on the costs and time differences of using the collar compared to behavioural observations from video recordings. Each hour of video took 1 to 3 h of human observer time to collect the data and input them into a spreadsheet. If a technician was being paid, this would be expensive as well as time-consuming. Thus, the annual cost of the collars and the possibility to save an enormous amount of time are, per se, advantages of choosing this device over human observations. However, there are some details of the behaviours that only a human observer can record, and this could be important for discussing the results.

## 5. Conclusions

Despite finding some significant differences in data produced by a wearable sensor collar in comparison with behavioural observations to measure sleep, we found merit in the use of this wearable technology. It should be remembered that even direct behavioural observations can vary significantly when different or even the same observers are involved, hence the use of interobserver reliability tests. Thus, comparisons between studies using different methods of data collection should be conducted with care.

## Figures and Tables

**Figure 1 animals-13-01467-f001:**
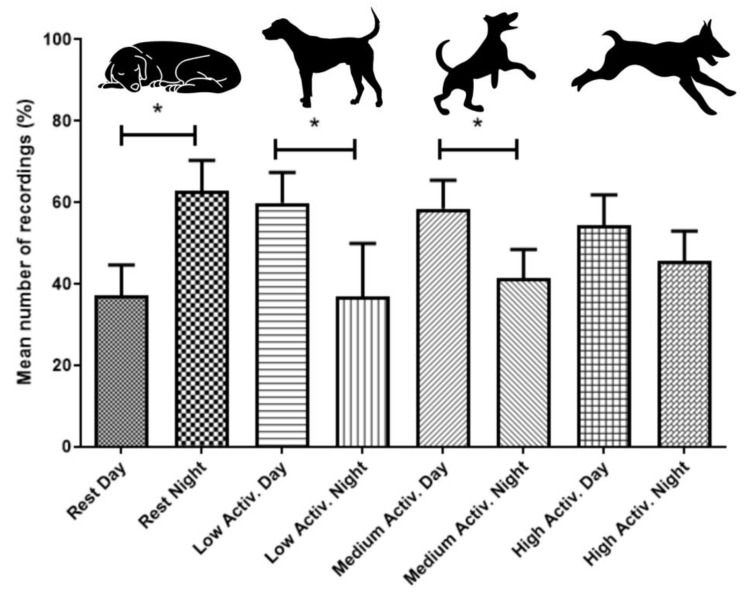
Mean expression of different activity levels (as percentages) of laboratory dogs as measured by PetPace collars over 20 days and nights per dog. Lines above columns with stars show significant differences between categories (*p* < 0.05). Error bars show ± standard error of the mean.

**Figure 2 animals-13-01467-f002:**
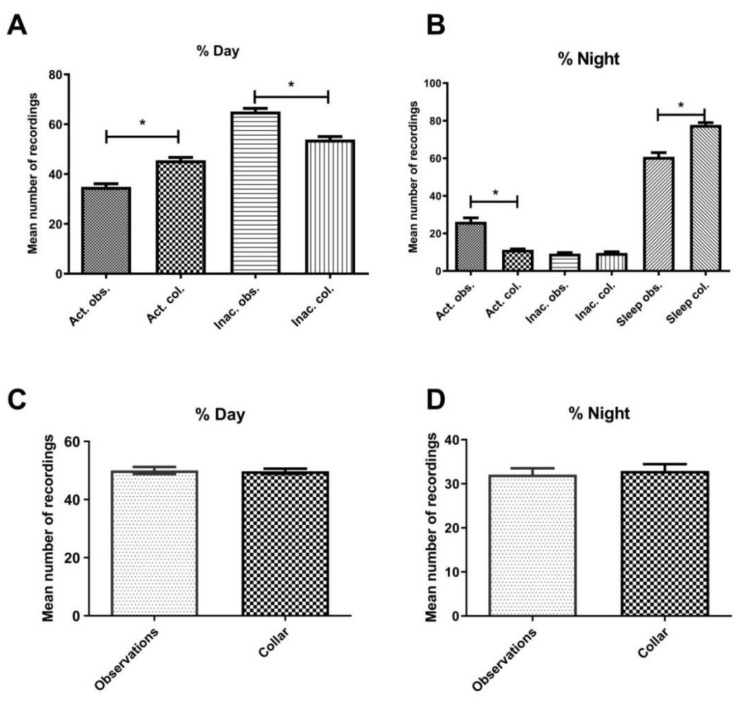
Comparison of behavioural recordings using two different methodologies: the PetPace wearable collar and behavioural observations from videos. (**A**) Mean expression of activity and inactivity during the day as recorded by different methods. (**B**) Mean expression of activity, inactivity, and sleep during the night, as recorded by different methods. (**C**) Total number of recordings for diurnal observations as registered by different methodologies. (**D**) Total number of recordings for nocturnal observations as registered by different methodologies. Lines above columns with stars show significant differences (*p* < 0.05). Error bars show ± standard error of the mean. Act = activity; Col = collars, Inac = Inactivity; Nig = night.

**Figure 3 animals-13-01467-f003:**
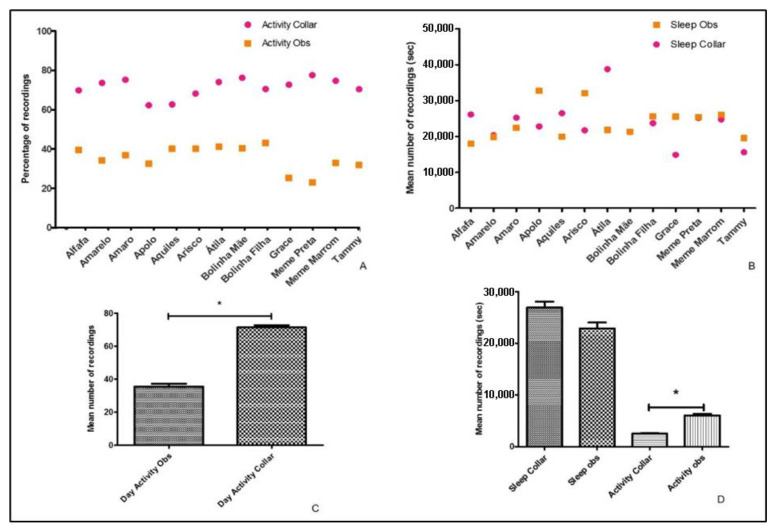
Comparison between automatic results generated by the PetPace collar activity threshold and the metrics recorded using video observations. (**A**) Individual variation of activity as recorded by different methods. (**B**) Individual variation of sleep behaviour as recorded by different methods. (**C**) Mean expression of diurnal activity as recorded by different methods. (**D**) Mean expression of activity and sleep during the night, as recorded by different methods. Lines above columns with stars show significant differences (*p* < 0.05). Error bars show ± standard error of the mean.

**Table 1 animals-13-01467-t001:** Inter-method differences for categories of behaviour measured in laboratory dogs.

Categories	Behaviour (±SD) ^1^	Collar (±SD) ^1^	Range ^2^	Difference ^3^
Activity Day	34.8% ± 15.9%	45.2% ± 15.6%	0.1–59.3%	10.5% *
Inactivity Day	65.2% ± 15.9%	53.5% ± 16.0%	1.4–75.5%	11.6% *
Activity Night	25.9% ± 26.5%	11.3% ± 6.0%	0.2–92.4%	14.6% *
Inactivity Night	9.1% ± 7.1%	9.6% ± 7.6	0.1–35.0%	6.4%
Sleep Night	60.0% ± 29.1%	77.8% − 14.6%	1.8–97.0%	17.8% *

^1^ Average observation. ^2^ Minimum and maximum values observed. ^3^ Mean difference for paired observations. * Statistical difference.

## Data Availability

All data are available at: Schork, Ivana; Manzo, Isabele; Beiral, Marcos; Costa, Fernanda; Young, Robert; Azevedo, Cristiano (2022). Using wearable technology to measure sleep behaviour and general physiology of domestic dogs—a prospective tool for animal welfare assessment. Mendeley Data, V1, doi: 10.17632/x768pztf5v.1.

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
