# Peer review of "Testing the Accuracy of Wearable Technology to Assess Sleep Behaviour in Domestic Dogs: A Prospective Tool for Animal Welfare Assessment in Kennels"

_animals, 2023, doi:10.3390/ani13091467_

Round 1

Reviewer 1 Report

REV Using wearable technology to measure sleep behaviour and general physiology of domestic dogs a prospective tool for animal welfare assessment.

The authors explored the usefulness of a new, non-invasive, methodology to measure sleep. They tested the accuracy of a smart-sensing collar from the brand PetPace™ to investigate sleep behaviour in domestic dogs. They performed a comparison between behavioural observation by video recording and associated physiological parameters such as heart and respiratory rate recorded by the sensors to behaviours.

The study is very interesting. Exploring the quantity and the quality of sleep in animals is an important indicator for the assessment of animal welfare, in particular the positive animal welfare. The use of non-invasive method such as sensors permits to overcomes the limits of data collection by conventional methodologies which require animal manipulation.

Some work is needed to improve the manuscript to be suitable for publication in this journal.

Please, consider the following comments.

LL 35-36 the physiological data recorded by the sensor were not compared with other method of assessment. So, this phrase could be ambiguous. I suggest to explain better this point. See also LL 130-131. It is not clear how the authors did assess the accuracy of the physiological parameters

LL 152-153 please specify the breed of dogs involved I the study as well as the weight. Regarding the age of dogs, the authors said that all dogs were adult but, in the results and discussion, they underlined that older dogs showed difference in activity. It could be useful explain better this point adding the age of the dogs considered older. Seen also LL 357.

LL 169-175 this part is not totally clear. Please clarify it explain better how the data collection was carried out. Which are the limitations?

LL 178-179 I suggest to add brief information regarding the behavioural assessment (Schork et al. is citation number 62?). The manuscript may benefit from it. Why is the observer? How many observers? Was an intra or inter-observer agreement performed? No supplementary material was available for the used ethogram, so it was not possible to understand the meaning of the chosen categories.

LL 216-218 as said before the values of physiological parameters recorded with sensor were not compared to the values recorded with other methods.

Figure 3 no results were available for rest/inactivity. Why?

LL 434 also sleep? In general, the use of rest and sleep appears confusing. Please verify this aspect.

LL 545-546 as said before, is it no clear if an intra or inter observer agreement was performed. The conclusion should be expanse particularly the merit found of this technology by the authors and also its limitations.

Author Response

Reviewer 1 Comments:

The authors explored the usefulness of a new, non-invasive, methodology to measure sleep. They tested the accuracy of a smart-sensing collar from the brand PetPace™ to investigate sleep behaviour in domestic dogs. They performed a comparison between behavioural observation by video recording and associated physiological parameters such as heart and respiratory rate recorded by the sensors to behaviours.

The study is very interesting. Exploring the quantity and the quality of sleep in animals is an important indicator for the assessment of animal welfare, in particular the positive animal welfare. The use of non-invasive method such as sensors permits to overcomes the limits of data collection by conventional methodologies which require animal manipulation.

Some work is needed to improve the manuscript to be suitable for publication in this journal.

Please, consider the following comments.

Thank you for taking the time to review our paper. Please see below the responses to the comments made.

LL 35-36 the physiological data recorded by the sensor were not compared with other method of assessment. So, this phrase could be ambiguous. I suggest explaining better this point. See also LL 130-131. It is not clear how the authors did assess the accuracy of the physiological parameters.

Thank you very much for your comment. Based on comments received from your review and to match suggestions made by the other reviewers, the authors have decided to remove the physiological data of the paper and focus only on the behavioural comparison. Please refer to the edited manuscript to see all relevant changes.

LL 152-153 please specify the breed of dogs involved I the study as well as the weight. Regarding the age of dogs, the authors said that all dogs were adult but, in the results and discussion, they underlined that older dogs showed difference in activity. It could be useful explain better this point adding the age of the dogs considered older. Seen also LL 357.

Thank you for your suggestion. Additional information on dogs’ breed, age range and weight has been added to Methods subsection – study site and subjects.

LL 169-175 this part is not totally clear. Please clarify it explain better how the data collection was carried out. Which are the limitations?

Thank you for your question. Further information has been added to improve clarity.

LL 178-179 I suggest to add brief information regarding the behavioural assessment (Schork et al. is citation number 62?). The manuscript may benefit from it. Why is the observer? How many observers? Was an intra or inter-observer agreement performed? No supplementary material was available for the used ethogram, so it was not possible to understand the meaning of the chosen categories.

Thank you for your suggestion. A full description of behavioural categories has been included in the supplementary material provided by the authors. Apologies if this file was not available to you. The reference to the previous publication includes more information on the camera's settings but not on the metrics evaluated; this should be evident from the supplementary content provided.

The behaviour recordings were conducted by a single observer using the recorded videos. No live observations were conducted, and as there was only one person collecting data, no interobserver reliability test was carried out.

LL 216-218 as said before the values of physiological parameters recorded with sensor were not compared to the values recorded with other methods.

This information has been removed, as explained in response to your first commentary.

Figure 3 no results were available for rest/inactivity. Why?

Thanks for your question. Figure 3 display results for the two additional categories created for evaluation of most displayed behaviours, as specified in the methods and reported in the results. 

LL 434 also sleep? In general, the use of rest and sleep appears confusing. Please verify this aspect.

Thank you for the suggestion, this has been clarified in the text.

LL 545-546 as said before, is it no clear if an intra or inter observer agreement was performed. The conclusion should be expanse particularly the merit found of this technology by the authors and also its limitations.

Thank you for the suggestion, although as explained, only a single observer performed the data collection for the behaviour.  

Reviewer 2 Report

Using wearable technology to measure sleep behaviour and general physiology of domestic dogs – a prospective tool for animal welfare assessment  

By  Schork et al.

The authors reported in this manuscript the second part of a research aimed to study interactions between sleep and behavior in order to evaluate whether sleep characteristics/quality can be useful indicators for dogs’ welfare. In this manuscript authors present data aimed to evaluate the validity of a wearable technology for automatic measuring of sleep, behavior and physiological parameters as HR and respiratory rate. The paper is well written and accurate. I have only one major point and very few comments/suggestions.

The major issues is the lack of interobservers reliability test for behavior scored from videos. Supplementary material reporting the dog ethogram is missing and I had no chance to check if the interobserver agreement was reported here. As the authors stated in their conclusion (lines 544-546) this is a fundamental point when analyzing behavior. Please add statistics for interobserver agreement in the text. 

Minor: a) the numbering of figures is wrong, Figure 1 is reported after figure 2 and 3; b) Schork et al.,  needs numbering [51] at line 178; c) sleep index at lines 236-243 is not straightforward, an example would be of help.

Author Response

Reviewer 2 comments:

The authors reported in this manuscript the second part of a research aimed to study interactions between sleep and behavior in order to evaluate whether sleep characteristics/quality can be useful indicators for dogs’ welfare. In this manuscript authors present data aimed to evaluate the validity of a wearable technology for automatic measuring of sleep, behavior and physiological parameters as HR and respiratory rate. The paper is well written and accurate. I have only one major point and very few comments/suggestions.

Thank you for taking the time to review our paper. Please see below the responses to the comments made. Please note, based on comments received from your review and to match suggestions made by the other reviewers, the authors have decided to remove the physiological data of the paper and focus only on the behavioural comparison. Please refer to the edited manuscript to see all relevant changes.

The major issues is the lack of interobservers reliability test for behavior scored from videos. Supplementary material reporting the dog ethogram is missing and I had no chance to check if the interobserver agreement was reported here. As the authors stated in their conclusion (lines 544-546) this is a fundamental point when analyzing behavior. Please add statistics for interobserver agreement in the text. 

Thank you for the comments. Inter-reliability test was not undertaken as there was a single observer conducting data collection for the behaviour.

Minor: a) the numbering of figures is wrong, Figure 1 is reported after figure 2 and 3;

Thank you for the observation, this has been corrected in the text.

  1. b) Schork et al.,  needs numbering [51] at line 178;

Thank you. Incorporated as suggested.

  1. c) sleep index at lines 236-243 is not straightforward, an example would be of help.

Thank you for the suggestion, an example has been included.

Reviewer 3 Report

Integrating technology into behaviour science is a fascinating field of research, and the manuscript addresses an interesting and timely question so I was excited to read it. Unfortunately, however, there are some fundamental problems with the paper.

One is insufficient problem statement. It would be important to clearly define and frame the question(s?) of the study.

Although the main aim seems to be to compare the coding efficiency of a human coder with a technical device, we basically learned very little about the smart-sensing collar being used to make sure that it actually measures what the manufacturer claims (especially related to “collecting through an acoustics sensor, physiological measures such as respiration rate and heart rate”).

It is not clear whether the paper attempts to validate the smart-sensing collar by using human observations as a control (whether the collar is as good as a human coder) or whether the device is assumed to be better (“Video analysis by a human observer is less will be as accurate in recording data as an electronic sensor."), but in this case, on what basis do they claim this?

Basically, the manuscript is very difficult to read and understand. This is largely due to the unclear objectives (which lead to unclear results and explanations) and the too many investigated questions (e.g., the activity pattern of dogs and the calculation of correlations between measured physiological data), which are not the main focus of the paper. It is also not clear why such obvious associations are analyzed that awake dogs have a higher heart rate and breathing rate than when they are asleep, and dogs are more active during the day than at night, etc. I would recommend a significant reduction in the number of questions/tests (especially since no statistical correction was applied despite the large number of tests).

Another serious issue is the quality of editing and English language. The manuscript would need extensive language and style editing. This is more than typos or minor mistakes, there are many hard-to-understand sentences, e.g.:

“The ability of the collar to measure behaviour had contradictory results.

“Another interesting result was the similar patterns of HR and RR variation observed in paired dogs. Although this difference did not have any statistical significance, it could be interesting…”

“Despite our attempts to collect similar data from both methodologies it is likely that a difference still occurred, which may partially explain our results. In summary, data from collars (just like direct behavioural observations) are precise but not necessarily accurate. “

It seems that the conclusion could well be the introduction: “wearable technology shows potential to be a useful, and a less-time consuming tool for the evaluation of behaviours and for the assessment of wellbeing in dogs”. I mean, a more critical evaluation of the results would be welcome. The authors state that “wearable technology tested was found to be a useful, and a less-time consuming tool in comparison to direct behavioural observations for the evaluation of behaviours and their connection to wellbeing in dogs”. These conclusions are not supported by the data, on the contrary, the results of the two methods significantly differed in almost all variables. This is the main result. To sum concrete behaviours until no difference can be found between the methods across the total diurnal and nocturnal observations does not appear to be a valid and sound approach. Instead, the authors should put more effort into figuring out what could be the reason for the significant differences.

Since the entire manuscript requires thorough editing and rewriting, I will not go into detail about the minor problems, will only mention a few:

Introduction

It is quite strange that, after much has been written about sleep in general and in various mammal species, the expanding literature on non-invasive sleep research related to family dogs is completely absent from the Introduction.

Methods

How could seven male adult dogs be kept in pairs?

Did the dogs really not sleep during the day?

Were the dogs relatives (littermates)?

Result

All descriptive data should be summarised in a table to make the text readable.

“dogs ate more when they had elevated heart rates” – not the other way around?

Discussion

The authors argue: “this feature (sleep fragmentation) in a combination index with heart rate and/or respiratory rate could provide relevant information about the different sleep phases”. To me, this seems highly unlikely. There is no reference added here, and apparently sleep phases were not analysed in the study.

There is no limitation section, although there were several limitations, for example, the measurements were performed on a specific population of lab dogs that were kept/kennelled/handled very differently than typical family dogs.

Author Response

Reviewer 3 comments:

Integrating technology into behaviour science is a fascinating field of research, and the manuscript addresses an interesting and timely question so I was excited to read it. Unfortunately, however, there are some fundamental problems with the paper.

One is insufficient problem statement. It would be important to clearly define and frame the question(s?) of the study.

Although the main aim seems to be to compare the coding efficiency of a human coder with a technical device, we basically learned very little about the smart-sensing collar being used to make sure that it actually measures what the manufacturer claims (especially related to “collecting through an acoustics sensor, physiological measures such as respiration rate and heart rate”).

It is not clear whether the paper attempts to validate the smart-sensing collar by using human observations as a control (whether the collar is as good as a human coder) or whether the device is assumed to be better (“Video analysis by a human observer is less will be as accurate in recording data as an electronic sensor."), but in this case, on what basis do they claim this?

Basically, the manuscript is very difficult to read and understand. This is largely due to the unclear objectives (which lead to unclear results and explanations) and the too many investigated questions (e.g., the activity pattern of dogs and the calculation of correlations between measured physiological data), which are not the main focus of the paper. It is also not clear why such obvious associations are analyzed that awake dogs have a higher heart rate and breathing rate than when they are asleep, and dogs are more active during the day than at night, etc. I would recommend a significant reduction in the number of questions/tests (especially since no statistical correction was applied despite the large number of tests).

Another serious issue is the quality of editing and English language. The manuscript would need extensive language and style editing. This is more than typos or minor mistakes, there are many hard-to-understand sentences, e.g.:

“The ability of the collar to measure behaviour had contradictory results.”

“Another interesting result was the similar patterns of HR and RR variation observed in paired dogs. Although this difference did not have any statistical significance, it could be interesting…”

“Despite our attempts to collect similar data from both methodologies it is likely that a difference still occurred, which may partially explain our results. In summary, data from collars (just like direct behavioural observations) are precise but not necessarily accurate. “

It seems that the conclusion could well be the introduction: “wearable technology shows potential to be a useful, and a less-time consuming tool for the evaluation of behaviours and for the assessment of wellbeing in dogs”. I mean, a more critical evaluation of the results would be welcome. The authors state that “wearable technology tested was found to be a useful, and a less-time consuming tool in comparison to direct behavioural observations for the evaluation of behaviours and their connection to wellbeing in dogs”. These conclusions are not supported by the data, on the contrary, the results of the two methods significantly differed in almost all variables. This is the main result. To sum concrete behaviours until no difference can be found between the methods across the total diurnal and nocturnal observations does not appear to be a valid and sound approach. Instead, the authors should put more effort into figuring out what could be the reason for the significant differences.

Since the entire manuscript requires thorough editing and rewriting, I will not go into detail about the minor problems, will only mention a few:

Introduction

It is quite strange that, after much has been written about sleep in general and in various mammal species, the expanding literature on non-invasive sleep research related to family dogs is completely absent from the Introduction.

Thank you very much for taking the time to revise the paper. Based on the suggestions, the authors have removed the physiological data altogether and restructured the paper to focus on comparing behavioural data, with emphasis on sleep, between the two methods.

A Native speaker has also revised the paper and made improvements to ensure clarity and good use of the English language.

We hope these amendments have produced a paper that better reflects the standard expected for publication in Animals and addressed all the previous points raised in your review.

Please see below additional responses for specific questions asked.  

Methods

How could seven male adult dogs be kept in pairs?

Thank you for your comment. All dogs initially were kept as pair (14 dogs in total), but during the pilot study one dog was removed from the research, hence why only 13 remained, this has been clarified in the methods subsection subjects and study site.

Did the dogs really not sleep during the day?

Thank you for your question. Unfortunately, sleeping during the day was rare, and only three individuals were recorded sleeping. This accountant for less than 0.5% of observations between all behaviours recorded during the day. The sleeping patterns of dogs have been thoroughly discussed in a previous publication - https://www.nature.com/articles/s41598-021-04502-2

Were the dogs relatives (littermates)? 

Thank you for the question. Aside from one female pair (mother-daughter) none of the other dogs were related.

Result

All descriptive data should be summarised in a table to make the text readable.

Thank you for your suggestion. Following the removal of the physiological data, the results section has significant less information and most results are reported as tables or graphs.  

“dogs ate more when they had elevated heart rates” – not the other way around?

Thank you for the initial question, no further considerations have been made on physiology following the removal of the data.

Discussion

The authors argue: “this feature (sleep fragmentation) in a combination index with heart rate and/or respiratory rate could provide relevant information about the different sleep phases”. To me, this seems highly unlikely. There is no reference added here, and apparently sleep phases were not analysed in the study.

Thank you for the comment. Response is similar to above.

There is no limitation section, although there were several limitations, for example, the measurements were performed on a specific population of lab dogs that were kept/kennelled/handled very differently than typical family dogs.

Thank you for the comment. Limitations have been addressed across discussion areas rather than in a specific section. The authors agreed that this would be a better way to point out specific areas for improvement or future directions in research.

Regarding the measurements mentioned, it is unclear if the comment refers to behaviour or physiology. Nonetheless, the idea was to initially investigate if the collar would be a suitable replacement for behavioural observations due to the time-consuming nature of the method, rather than establish its many applications. A setting with less variation and having a population with less interference as possible was perceived as more suitable in this case. Furthermore, The collar is already sold and believed to be a reliable device  (from previous  studies), therefore the scope of the research was merely check if sleep was also accurate, hence why the discussion has not explored the further applications limitations of its use.